# Dynamic Weighting: Exploiting the Potential of a Single Weight Across Different Modes

## Abstract

Weights play an essential role in determining the performance of deep networks. This paper introduces a new concept termed "Weight Augmentation strategy"(WAS), which emphasizes the exploration of weight spaces rather than traditional network structure design. The core of WAS is the utilization of randomly transformed weight coefficients, referred to as Shadow Weights (SW), for deep networks to calculate the loss function and update the parameters. Differently, stochastic gradient descent is applied to Plain Weights (PW), which is referred to as the original weight of the network before the random transformation. During training, numerous SW collectively form a high-dimensional space, while PW is directly learned from the distribution of SW. To maximize the benefits of WAS, we introduce two operational modes, *i.e.*, the Accuracy-Priented Mode (AOM) and the Desire-Oriented Mode (DOM). To be concrete, AOM relies on PW, which ensures that the network remains highly robust and accurate. Meanwhile, DOM utilizes SW, which is determined by the specific objective of our proposed WAS, such as reduced computational complexity or lower sensitivity to particular data. These dual modes can be switched at any time as needed, thereby providing flexibility and adaptability to different tasks. By extending the concept of augmentation from data to weights, our WAS offers an easy-to-understand and implement technique that can significantly enhance almost all networks. Our experimental results demonstrate that convolutional neural networks, including VGG-16, ResNet-18, ResNet-34, GoogleNet, MobileNetV2, and EfficientNet-Lite, benefit substantially with little to no additional costs. On the CIFAR-100 and CIFAR-10 datasets, model accuracy increases by an average of 7.32% and 9.28%, respectively, with the highest improvements reaching 13.42% and 18.93%. In addition, DOM can reduce floating point operations (FLOPs) by up to 36.33%.

## 1 Introduction

learning with data augmentation (DA) has achieved great success, which significantly enhances model performance through various preprocessing methods such as rotation, translation, scaling, and random cropping Moreno-Barea et al. (2020); Shorten et al. (2021); Yang et al. (2022).The primary goal of DA is to align the distribution of the original dataset more closely with that of natural scenes, thereby increasing both the diversity and quantity of data. Wang *et al.* Hao & Zhili (2020) advanced this field by improving the Mosaic data augmentation algorithm, which analyzes synthetic image areas and randomly fills them with a certain number of training images. Additionally, researchers at CMU Trabucco et al. (2023) proposed the DA-Fusion strategy using pre-trained text-to-image diffusion models to generate diverse variants of real images, thereby enhancing data variety and improving model robustness.

Although complex DA techniques can enhance model accuracy, they come with notable drawbacks He et al. (2019); Dosovitskiy et al. (2020); Cubuk et al. (2020); Tian et al. (2020); Zoph et al. (2020). First, intricate augmentation strategies can make models difficult to implement and customize. Second, it is challenging to avoid generating irrelevant tasks or misleading augmented data. For example, issues such as data over-enhancement, where the model cannot effectively learn from the augmented data, and deviations of the augmented data from the original distribution can arise Suresh et al. (2021); Zheng et al. (2024); Wang et al. (2022); Jiang et al. (2020); Gou et al. (2021).

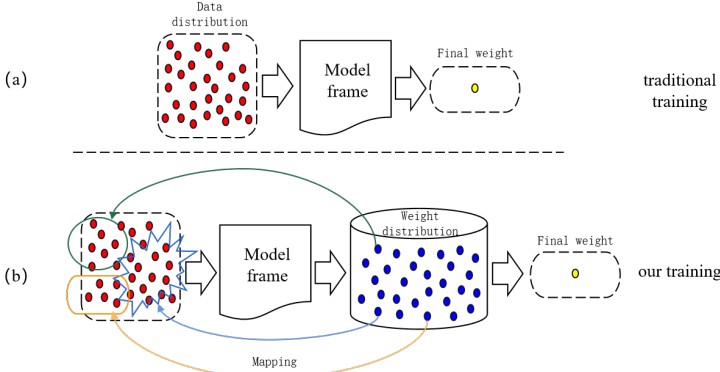

Figure 1: This figure contrasts traditional and innovative model training strategies. Panel (a) depicts the traditional method where specific parameters are derived directly from the data, represented by red circles. In contrast, panel (b) introduces our novel strategy where the distribution of weights, rather than the weights themselves, is learned from the data.

Weights are the core part of model deployment and the basis for decision-making in practical applications, which do not directly interfere with the distribution of real data Zamfirescu-Pereira et al. (2023); Alberts et al. (2023); Ghosal et al. (2023). Since the advent of deep learning, the dominant approach to model training has been to utilize data to optimize model parameters, aiming to obtain a set of weights that lead to superior performance. However, the outcomes are not always favorable. Izmailov *et al.* Izmailov et al. (2018) proposed stochastic weight averaging, which integrates the weights generated during the training stage by averaging multiple weights, thereby abandoning the final weight result. In traditional training methods, inference relies on a single weight, which can be directly obtained. However, identifying a set of parameters from the weight space that satisfies the feature space mapping is challenging, particularly when both the weight space and the feature space are high-dimensional. Figure 1(a) illustrates the traditional training method, where the relationship between data and weights is many-to-one. Model training aims to find weights that correspond to the data. The formula can be expressed as follows:

$$\Phi : \mathbb{R}^m \to x \tag{1}$$

In this context, $\Phi$ represents the function that maps an $m$-dimensional feature space to the corresponding weights. Specifically, $\mathbb{R}^m$ is defined as the $m$-dimensional Euclidean space representing the feature vectors. Likewise, $\mathbb{R}^n$ denotes the $n$-dimensional space of potential weight vectors, illustrating the multidimensional space where the weights reside. Figure 1(b) illustrates our innovative strategy, where the relationship between data and weights is conceptualized as many-to-many. Throughout the training process, the weight space becomes increasingly complete. With the application of the proposed WAS, the weight space will be further expanded. Instead of seeking the optimal weights directly, our approach involves learning the distribution of the entire weight space. The relationship is mathematically expressed as follows:

$$\Psi : \mathbb{R}^m \to \mathbb{R}^n \tag{2}$$

where $\Psi$ denotes the transformation that maps the $m$-dimensional feature space to the $n$-dimensional weight space. To address the shortcomings of traditional methods like Dropout Srivastava et al. (2014), which combats overfitting by randomly dropping units during training but does not reduce computational complexity in inference, we introduce Weight Augmentation Strategy (WAS). This technology is influenced by both dropout and data augmentation strategies and takes a novel approach by randomly transforming weights during the training phase. These transformed weights, referred to as SW, are employed to compute the loss function and facilitate the update of model parameters. In contrast, during the inference phase, Stochastic Gradient Descent (SGD) is applied to PW. This dual-weight strategy ensures that the model achieves optimal performance only when PW aligns effectively with the majority of SW. As a result, WAS allows for a more robust system that improves upon the limitations of dropout by considering the computational demands during inference.

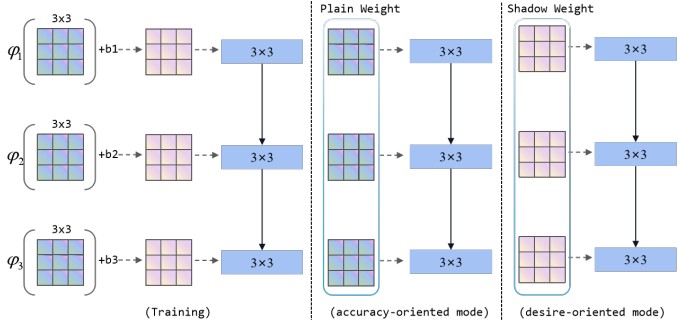

Figure 2: Sketch of WAS architecture. There exist two modes during inference, namely the accuracy-oriented mode using PW and the desire-oriented mode using SW.

While our system utilizes two different types of weights, only the PW need to be saved. The SW are generated through random transformations applied to PW. This approach allows us to maintain a single set of weights that can operate in two distinct modes, which offers different functionalities at minimal cost. Depending on the task requirements, the operational mode can be switched dynamically. As depicted in Figure 2, AOM utilizes PW to ensure high robustness and optimal handling of most data types. In contrast, DOM employs SW, which can be tailored for specific functions, such as reducing computational complexity and enhancing sensitivity to particular data types. Our contributions can be summarized as follows:

- We propose WAS, which enhances the robustness of model training through the random transformation of weights.

- The concept of dual-mode weights allows for retaining just one set of model weights during inference, enabling the model to adapt to multiple states and meet various task requirements efficiently.

- Our approach redefines traditional training by focusing on the entire distribution of weights rather than optimizing for a specific weight, with the goal of identifying the most effective and robust weight configuration for diverse applications.

## 2 RELATED WORK

With the continuous development of deep learning, researchers have begun to seek more efficient and economical methods to obtain model weights. WiSE-FT Wortsman et al. (2022) enhances the robustness of fine-tuned pre-trained models by integrating the weights of zero-shot and fine-tuned models. This method can maintain high accuracy and adapts well to changes in data distribution. However, the performance of WiSE-FT largely depends on the pre-trained model and fine-tuning data. Guo *et al.* Guo et al. (2020) introduced a collaborative knowledge distillation approach that trains multiple student models simultaneously. This strategy enhances learning efficacy without the need for a separate teacher model, though the selection and quantity of student models impact the outcome and require substantial computational resources. Zhang *et al.* Zhang et al. (2019) developed a self distillation technique that employs the network as both the student and the teacher, facilitating internal knowledge transfer. It does not require an additional pre-trained teacher model and can improve accuracy without increasing inference time. However, self distillation introduces an additional shallow classifier, which prevents the model convergence and increases the complexity of training.

To overcome the limitations of these methods, we introduce the Weight Augmentation Strategy (WAS). We randomly transform PW during training to obtain SW. PW is used to be compatible with various SW, in order to enhance the robustness of the network and reduce the sensitivity of the model to noise.

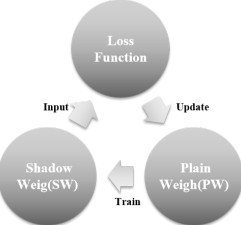

Figure 3: Interaction Triangle of WAS Components. SW are utilized to compute the loss function, which influences the adjustment of PW.

## 3 METHOD

### 3.1 WEIGHT AUGMENTATION STRATEGY

In traditional training, it is a common occurrence that many weights generated are considered "by-products" and typically disregarded as they are not optimal for broader data applications, despite their effectiveness on specific datasets. These weights are usually discarded during the model selection process. However, with the deepening of deep network research, the potential value of these "by-products" has been recognized. For example, ensemble learning methods Dietterich et al. (2002); Krawczyk et al. (2017); Huang et al. (2009) can integrate these seemingly useless weights to form a more powerful and robust model. These conventional approaches of leveraging diverse weights have not fully tapped into the potential of weight augmentation. A robust weight space, where weights are learned from distributions rather than specific data points, can significantly improve performance. The augmentation involves transforming weights through various methods such as rotation, translation, and scaling, as shown in the equation:

$$h\left(x\right) = ReLU\left(TW^{'}x\right) = ReLU\left(\sum_{i=1}^{k} T_i W_i x_i + b\right)$$

(3)

where $W$ and $W^{'}$ denote the weight vector and its transposition, respectively, and $T_i$ represents the transformation applied to each weight component.

However, merely reorganizing weights does not fully capture the weight distribution and can lead to biases and reduced generalization ability. To address the above issues, we introduce randomness into the augmentation process, thereby enhancing the diversity and representativeness of the weight space:

$$h\left(x\right) = ReLU\left(\sum_{i=1}^{j} \gamma\left(T_i\right) W_i x_i + b\right),$$

(4)

where $\gamma\left(T_i\right)$ controls the degree of randomness in the transformation, which is crucial for optimizing the efficacy of WAS. The application of randomness is adjusted based on the requirements of convolutional layers and the specific augmentation strategy chosen, such as the degree of translation or scaling. By integrating these elements, WAS aims to refine the way weights are used and adjusted, promoting a more dynamic and flexible approach to deep learning model training.

WAS is an innovative strategy designed to optimize deep networks by directly applying randomness and constraints to weights, thus enhancing the model's capability to explore the weight space. This approach not only addresses the limitations faced by traditional weight optimization methods, such as a lack of flexibility and autonomy, but also revitalizes the training process by introducing dynamic changes in weight configurations.

As training continues, the distribution of SW is refined, enhancing the overall model parameters. The updating of model parameters is governed by the following equations:

$$\theta_j = \theta_j - \alpha \cdot \frac{\partial}{\partial \theta_j} J(\theta)$$

(5)

$$\theta_j^{pw} = \theta_j^{pw} - \alpha \cdot \frac{\partial}{\partial \theta_j^{pw}} J(\theta_j^{sw})$$

(6)

Table 1: Deep networks defined by WAS

| Name | Crop | Translate | Rotate |
|------|------|-----------|--------|
| Model-C | $\checkmark$ | | |
| Model-T | | $\checkmark$ | |
| Model-R | | | $\checkmark$ |
| Model-CT | $\checkmark$ | $\checkmark$ | |
| Model-CTR | $\checkmark$ | $\checkmark$ | $\checkmark$ |

Here, $\theta_j^{sw}$ and $\theta_j^{pw}$ represent the $j$-th parameters of SW and PW, respectively. $\alpha$ denotes the learning rate and $J(\theta)$ is the loss function. The derivative $\frac{\partial}{\partial \theta_j^{sw}}$ reflects the partial derivative of the loss function $J(\theta_j^{sw})$ with respect to the parameter $\theta_j^{pw}$.

Additionally, the interaction between SW and PW within the training process is captured by

$$\frac{\partial J}{\partial \theta_j} = \sum_{x,y} \frac{\partial J}{\partial Z_{xy}} \cdots \frac{\partial Z_{xy}}{\partial \theta_j}, \tag{7}$$

where $Z_{xy}$ represents the feature map at position (x, y). The partial derivatives $\frac{\partial J}{\partial Z_{xy}}$ and $\frac{\partial Z_{xy}}{\partial \theta_j}$ quantify the sensitivity of the loss function to changes in the feature map and the feature map's responsiveness to changes in the weights, respectively. Here, SW is employed to compute the loss function, which assesses the discrepancy between the model's outputs and the actual results. These computations not only impact PW adjustments but also facilitate learning the data distribution via SW. Thus, PW is primarily updated based on the distribution of SW, which is a significant departure from conventional methods where direct data-driven updates are commonplace. The relationship between PW, SW, and loss function is shown in Figure 3. The WAS incentive model integrates hundreds of weight training iterations, leveraging the advantages of SW while mitigating potential drawbacks. This strategy enhances model performance and allows for mode switching based on task requirements, paving the way for more intelligent and efficient deep network training.

In related research, Ding *et al.* Ding et al. (2021) introduced RepVGG, a method that adjusts weights during inference to reduce time complexity while preserving performance. This approach explores the potential relationship between model structures and weights, although it does not directly link the two. Meanwhile, Zheng *et al.* Zheng et al. (2023) introduced the "Learn From Model" concept, emphasizing the need for deeper exploration and modification of foundation models based on model interfaces.

Our approach takes a step further by advocating for weight-based training over traditional data-driven methods. By focusing on weight distributions rather than direct data features, our method allows the model to learn from a broader array of potential configurations, encompassing even flawed weights that can still capture specific data nuances. This two-stage training process—alternating between learning from data distributions and flawed weight distributions—leads to a model that optimizes its structure and weights autonomously without human intervention.

WAS not only encourages exploration of a wider weight space to find high-performance configurations but also enhances computational efficiency. This reduces resource consumption during both training and inference. Moreover, WAS can be tailored to specific model functions, such as processing particular types of data or reducing computational complexity, which aligns with the unique needs of various applications.

## 3.2 DUAL WORKING MODE OF NETWORK MODEL

The core of WAS is to encourage the generation of a large number of weights, which enriches the weight distribution and enhances the generalization and robustness of the model. The model operates in two distinct modes during training: AOM and DOM.

AOM operates similarly to traditional deep learning models and is utilized directly in inference. The innovation in AOM lies in its dynamic weight formation mechanism, which emerges through a competitive process during training. This competition drives continual optimization of the model's

Table 2: Comparison of top-1 accuracy (%) and average FLOPs (M) on CIFAR10 and CIFAR100 datasets

| Model | CIFAR10 | | CIFAR100 | | Average FLOPs (M) |
|---|---|---|---|---|---|
| | AOM Top1 | DOM Top1 | AOM Top1 | GOM Top1 | |
| **VGG** | 87.42 | - | 59.01 | - | 333 |
| VGG16-C | 89.29 | 88.95 | 62.19 | 60.53 | 321 |
| VGG16-CT | 91.15 | 90.60 | 63.53 | 61.96 | 279 |
| **ResNet18** | 85.53 | - | 59.70 | - | 608 |
| ResNet18-C | 88.15 | 87.38 | 63.42 | 62.44 | 542 |
| ResNet18-CT | 89.83 | 88.64 | 63.40 | 61.50 | 315 |
| **ResNet34** | 86.54 | - | 58.64 | - | 1214 |
| ResNet34-C | 89.18 | 88.81 | 62.19 | 61.24 | 1099 |
| ResNet34-CT | 91.71 | 90.73 | 60.64 | 58.73 | 709 |
| **GoogleNet** | 90.55 | - | 72.04 | - | 1457 |
| GoogleNet-C | 91.99 | 91.43 | 73.05 | 72.02 | 1237 |
| GoogleNet-CT | 92.68 | 92.04 | 73.08 | 72.01 | 772 |
| **MobileNetV2** | 73.2 | - | 47.02 | - | 47 |
| MobileNetV2-C | 82.58 | 82.43 | 55.43 | 54.65 | 45 |
| MobileNetV2-CT | 83.02 | 82.88 | 55.92 | 53.95 | 37 |
| **EfficientNetLite** | 73.05 | - | 43.13 | - | 8.00 |
| EfficientNetLite-C | 80.96 | 79.97 | 49.08 | 48.47 | 6.37 |
| EfficientNetLite-CT | 83.72 | 81.23 | 50.31 | 47.41 | 4.66 |

performance. Within this framework, SW that demonstrate superior performance can significantly reduce the loss function, thereby influencing the determination of PW. Weights that perform poorly exert less influence on the final weight determination.

DOM is specifically tailored to meet unique operational requirements such as enhancing model specificity, sparsity, and computational efficiency. In DOM, we can devise WAS tailored to the particular demands of the task and implement it within training. This adaptive strategy allows for the tuning of PW for use during inference. Although DOM may result in lower prediction accuracy compared to AOM, it offers substantial benefits for specialized tasks. For example, implementing strategies like random weight cropping can increase the sparsity of the weight matrix, thereby drastically reducing the computational load—sometimes by several orders of magnitude. This reduction is particularly valuable in scenarios involving large-scale data processing or in resource-constrained environments.

As a consequence, WAS provides a flexible model management strategy that only requires the storage of one set of weights while enabling two distinct functional modes. During inference, the model can switch between these modes based on the specific requirements of the application scenario. This dual-mode operation not only simplifies the management of model storage and maintenance but also enhances the algorithm's resilience and adaptability to varying conditions.

## 4 EXPERIMENTS

To evaluate the effectiveness of the proposed WAS, we conducted a series of ablation studies and comparative analyses on the CIFAR-10 and CIFAR-100 datasets Krizhevsky et al. (2009). The main objective of these experiments was to demonstrate the impact that WAS integration has on the models. Furthermore, we evaluate to delineate the distinctions and performance implications of dual operational modes. The results from these studies consistently confirm the advantages of incorporating WAS, showcasing substantial enhancements in model performance across various metrics.

Table 3: Comparison of AOM and DOM Top-1 Drop Rates under Different WAS

| Model | Data Augmentation | Parameters | AOM Top-1 Drop Rate(%) | GOM Top-1 Drop Rate(%) |
|---|---|---|---|---|
| **VGG16-R** $(0°, 90°)$ | rotate | $(0°, 15°)$ | 5.09 | 4.86 |
| | | $(0°, 45°)$ | 39.52 | 25.58 |
| | | $(0°, 90°)$ | 46.29 | 42.74 |
| | | $(0°, 135°)$ | 54.61 | 53.65 |
| | | $(0°, 180°)$ | 58.40 | 57.76 |
| **VGG16-T** (30%,30%) | translate | (10%,10%) | 1.72 | 2.33 |
| | | (20%,20%) | 6.62 | 8.47 |
| | | (30%,30%) | 14.42 | 18.21 |
| | | (40%,40%) | 25.76 | 31.76 |
| **VGG16-C** (0.8,1.0) | crop | (0.8,1.0) | 3.34 | 2.79 |
| | | (0.6,1.0) | 6.49 | 6.71 |
| | | (0.4,1.0) | 13.04 | 12.12 |
| | | (0.2,1.0) | 24.19 | 22.42 |

## 4.1 WAS FOR CLASSIFICATION

To evaluate the effectiveness of WAS, we have chosen six prominent deep learning architectures as our experimental models: VGG-16 Simonyan & Zisserman (2014), ResNet18 He et al. (2016), ResNet34 He et al. (2016), GoogLeNet Szegedy et al. (2015), EfficientNet-Lite Tan & Le (2019), and MobileNetV2 Sandler et al. (2018). The results of these comparative experiments are detailed in Table 1, showing improvements against baseline models.

We deliberately eschewed the incorporation of additional techniques, primarily to mitigate the impact of extraneous variables. On a solitary GPU, we established a global batch size of 128. We employed the conventional SGD, initializing the learning rate at 0.01. Furthermore, we fine-tuned the SGD optimizer, assigning a momentum coefficient of 0.9, thereby augmenting the model's stability and hastening convergence throughout the training regimen.

Table 2 demonstrates the performance improvement achieved by Model-C and Model-CT under different architectures. On the CIFAR-10 dataset, AOG for the models demonstrated discernible enhancements. Specifically, the accuracy of VGG16-C and VGG16-CT saw an increase of 2.13% and 4.27% relative to the baseline VGG16, respectively. For ResNet18, ResNet18-C and ResNet18-CT attained an accuracy improvement of 3.06% and 5.03%. Similarly, the ResNet34-C and ResNet34-CT models realized respective accuracy improvements of 3.05% and 5.97%. The GoogleNet-C and GoogleNet-CT models recorded accuracy enhancements of 1.59% and 2.35%. Within the MobileNetV2 series, the accuracy for MobileNetV2-C and MobileNetV2-CT marked a significant rise of 12.76% and 13.42% over MobileNetV2. Lastly, the EfficientNetLite-C and EfficientNetLite-CT models secured accuracy improvements of 9.52% and 13.23%.

On the CIFAR-100 dataset, WAS also demonstrated enhanced performance across various models. The VGG16-C and VGG16-CT models showed accuracy improvements of 5.39% and 7.66%, respectively.The ResNet18-C and ResNet18-CT models recorded increases of 6.23% and 6.19%, while the ResNet34-C and ResNet34-CT achieved accuracy gains of 6.05% and 3.41%, respectively. GoogleNet-C and GoogleNet-CT noted slight improvements with gains of 1.40% and 1.44%. Remarkably, MobileNetV2-C and MobileNetV2-CT marked significant advancements with increases of 17.89% and 18.93%. Finally, the EfficientNetLite-C and EfficientNetLite-CT models displayed notable accuracy enhancements of 13.78% and 18.93%, respectively, illustrating the substantial impact of WAS on model performance.

The performance of Model-C and Model-CT under DOM is slightly lower than under AOM, both models still outperform their baseline by approximately 1% to 2%. This indicates that both operational modes enhance model performance, with particularly notable gains in lightweight models. Furthermore, while not the primary focus, it is important to acknowledge the reductions in compu-

Table 4: Comparison of AOM and DOMTop-1 drop rates of different WAS under different data enhancements

| Model | Data Augmentation | Parameters | AOM Top-1 Drop Rate(%) | DOM Top-1 Drop Rate(%) |
|---|---|---|---|---|
| VGG16-R (0°,90°) | corp | (0.8,1.0) | 2.33 | 2.50 |
| | | (0.6,1.0) | 5.93 | 5.15 |
| | | (0.4,1.0) | 11.35 | 11.70 |
| | | (0.2,1.0) | 22.31 | 23.08 |
| | translate | (10%,10%) | 2.56 | 3.27 |
| | | (20%,20%) | 9.96 | 10.39 |
| | | (30%,30%) | 19.95 | 20.45 |
| | | (40%,40%) | 32.45 | 32.71 |
| VGG16-T (30%,30%) | rotate | (0°,15°) | 5.09 | 5.27 |
| | | (0°,45°) | 28.39 | 27.97 |
| | | (0°,90°) | 46.80 | 47.28 |
| | | (0°,135°) | 55.18 | 56.28 |
| | | (0°,180°) | 58.96 | 59.67 |
| | crop | (0.8,1.0) | 2.69 | 2.61 |
| | | (0.6,1.0) | 5.16 | 6.11 |
| | | (0.4,1.0) | 10.12 | 12.18 |
| | | (0.2,1.0) | 21.53 | 25.10 |
| VGG16-C (0.8,1.0) | rotate | (0°,15°) | 4.80 | 4.85 |
| | | (0°,45°) | 26.68 | 28.14 |
| | | (0°,90°) | 46.10 | 45.59 |
| | | (0°,135°) | 54.84 | 53.04 |
| | | (0°,180°) | 58.98 | 56.51 |
| | translate | (10%,10%) | 2.56 | 3.24 |
| | | (20%,20%) | 8.50 | 9.87 |
| | | (30%,30%) | 17.70 | 18.21 |
| | | (40%,40%) | 28.81 | 33.70 |

tational load as measured by FLOPs. For Model-C, FLOPs decreased by about 5% to 20%, whereas Model-CT saw a more substantial reduction, with FLOPs decreasing by up to 47%.

## 4.2 CHARACTERISTIC OF WAS

WAS endows models with unique capabilities that can be tailored to specific needs during inference. These capabilities encompass a range of enhancements, including reduced computational complexity and decreased sensitivity to particular types of data. This functionality is achieved by switching to SW during inference, allowing for customizable adjustments according to specific requirements. Consequently, WAS enables the model to adapt effectively to unique operational environments.

Table 3 showcases three representative WAS strategies implemented following the official PyTorch example Imambi et al. (2021): rotation (random rotation angles set between 0° and 90°), translation (random translation of weights up to 30% in horizontal and vertical directions), and cropping (random cropping of weights with ratios between 0.8 and 1.0). To assess the effectiveness of these strategies for processing specific data, we conducted random data augmentation on the test set.

After using WAS, the network operates in two modes: AOM, which does not apply WAS during inference, and DOM, which applies WAS during inference. As detailed in Table 3, when employing a random rotation strategy for WAS training (0° to 90°), AOM consistently outperforms DOM in Top-1 accuracy. Notably, with rotations limited to 0° to 45° and 0° to 90°, the accuracy losses in DOM compared to AOM are significantly reduced by 13.84% and 3.55%, respectively. For other rotation angles, the accuracy loss in DOM is approximately 1% lower than in AOM.

Additionally, when implementing the cropping strategy, as the weight cropping ratio increases, the accuracy loss in DOM compared to AOM decreases progressively by 0.58%, 0.22%, 0.92%, and

Table 5: Comparison of the impact of Top-1 drop rates of different randomly cropped WAS on rotation data

| WAS Strategy | AOM Top-1 acc | DOM Top-1 acc | Parameters | AOM Top-1 Drop Rate(%) | DOM Top-1 Drop Rate(%) |
|---|---|---|---|---|---|
| Crop (0.8,1.0) | 89.72 | 88.95 | (0°,15°) | 4.80 | 4.85 |
| | | | (0°,45°) | 26.68 | 28.14 |
| | | | (0°,90°) | 46.10 | 45.59 |
| | | | (0°,135°) | 54.84 | 53.04 |
| | | | (0°,180°) | 58.98 | 56.51 |
| Crop (0.6,0.8) | 90.34 | 89.71 | (0°,15°) | 5.58 | 5.04 |
| | | | (0°,45°) | 27.85 | 26.85 |
| | | | (0°,90°) | 46.54 | 45.82 |
| | | | (0°,135°) | 56.18 | 53.37 |
| | | | (0°,180°) | 59.98 | 56.16 |
| Crop (0.4,0.6) | 88.54 | 87.99 | (0°,15°) | 5.79 | 4.51 |
| | | | (0°,45°) | 27.44 | 24.62 |
| | | | (0°,90°) | 45.67 | 43.25 |
| | | | (0°,135°) | 54.66 | 51.57 |
| | | | (0°,180°) | 58.01 | 53.99 |
| Crop (0.2,0.4) | 87.09 | 86.19 | (0°,15°) | 5.96 | 4.43 |
| | | | (0°,45°) | 27.57 | 25.72 |
| | | | (0°,90°) | 45.85 | 44.4 |
| | | | (0°,135°) | 54.52 | 52.77 |
| | | | (0°,180°) | 58.01 | 54.88 |

1.77%. This indicates that the disparity in accuracy loss between DOM and AOM first narrows and then widens, reflecting the impact of increasing random crop ratios during training, which can degrade model performance if parameters exceed certain thresholds.

In summary, WAS helps to promote the ability to process specific data. However, when WAS involves random translation, the accuracy loss in DOM exceeds that in AOM, contradicting initial expectations. Upon integrating random cropping with translation, it appears that the reduction in model parameters may diminish the fitting capabilities of the model.

As depicted in Table 4, evaluated the effects of random translations within the test dataset through our AD pipeline, which deviates from traditional AD approaches. As the random translation parameter increased from 10% to 40%, the Top-1 accuracy drop rate in AOM escalated from 2.56% to 32.45%. Concurrently, DOM experienced a rise in the Top-1 drop rate from 3.27% to 33.70%. As the cropping ratio decreased from 1.0 to 0.2, both modes saw an increase in the Top-1 accuracy drop rate. For AOM, the drop rate increased from 2.69% to 21.53%,while for DOM, it rose from 2.61% to 25.10%.Notably, DOM exhibited a more pronounced performance decline, especially at higher cropping ratios. Based on the preliminary analysis of the above data, we can conclude that AOM has stronger generalization when the WAS strategy is inconsistent with the data augmentation strategy.

Similarly, as the rotation angle of the test set increases from 0° to 180°, using a random translation WAS strategy, the drop rate for AOM escalates from 5.09% to 58.96%. In DOM, the drop rate rises from 5.27% to 59.67%. Conversely, when employing a random cropping WAS strategy, the drop rate for AOM increases from 4.80% to 58.98%, while for DOM, it climbs from 4.85% to 56.51%. Under extreme augmentation conditions, the drop rate in AOM is unexpectedly higher than in DOM, which is contrary to initial expectations. This indicates that DOM may handle extreme data manipulations more robustly than AOM.

To explore the above phenomenon, Table 5 presents the models trained with varying random cropping ratios for WAS, while DA employs random rotation. On the test set, despite changes in training parameters, the drop rate of AOM remains relatively stable. However, as the random cropping ratio in WAS increases, the Top-1 accuracy drop rates for DOM are 2.47%, 3.82%, and 4.02% lower than those for AOM respectively, which suggests that DOM is more adept at processing rotated data.

Table 6: Training results on CIFAR10 datasets using different WAS

| Cropping parameters | Translation parameters | Mode Two Acc | FLOPs (M) | Average sparsity rate(%) |
|---|---|---|---|---|
| (1.0,1.0) | - | 87.42 | 333 | - |
| (0.8,1.0) | - | 88.95 | 320.68 | 3.70 |
| (0.6,0.8) | - | 89.71 | 259.67 | 22.02 |
| (0.4,0.6) | - | 87.99 | 223.08 | 33.01 |
| (0.2,0.4) | - | 86.19 | 212.02 | 36.33 |
| - | (30%,30%) | 90.37 | 230.63 | 30.74 |
| (0.8,1.0) | (30%,30%) | 90.60 | 277.42 | 16.69 |

As cropping ratios range from 0.2 to 0.4, the disparity in Top-1 accuracy drop rates between AOM and DOM narrows, which is likely due to a decrease in parameters which affects the model's ability to fit data. This underscores how data rotation, similar to cropping, can lead to information loss, yet DOM (employing weight-based random cropping) exhibits a lower drop rate than AOM, highlighting its robustness.

Further analysis of the model's structural efficiency is shown in Table 6. When cropping parameters are set at (0.4,0.6), the model achieves a sparsity rate of 33.01% with zero elements and maintains an accuracy of 87.99%, which is comparable to the base model. When the Cropping parameters are reduced to (0.2,0.6), the accuracy decreases by 1.41%, but the proportion of 0 elements increases by 3.32%, and the FLOPs are reduced by 11.06M. When introducing translation parameters (30%,30%), even without cropping, the sparsity rate is increased to 30.74%, floating-point operations(FLOPs) are reduced to 230.63M, and the accuracy is improved by 3.26% compared to the base model. Combining cropping with translation through WAS not only enhances model accuracy but also significantly lowers computational costs, demonstrating the dual benefits of this approach in improving model performance and efficiency.

## 5 CONCLUSION

We introduce WAS as a training method for deep learning models. Central to WAS is the implementation of a dual-mode inference system, which allows the weights to be tailored to meet diverse task requirements effectively. This approach facilitates the fine-tuning of weights to address specific needs precisely. In AOM, WAS has demonstrated the capability to enhance model accuracy by as much as 18.93% without additional computational expenses. In DOM, it can reduce FLOPs by up to 36.33%, while maintaining robust accuracy levels.

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

## A  APPENDIX

You may include other additional sections here.

