# OpenReview forum: "Dynamic Weighting: Exploiting the Potential of a Single Weight Across Different Modes"
_ICLR.cc/2025/Conference — Submitted to ICLR 2025_

### Official Review · Reviewer_h9nL · 2024-10-27

**Soundness:** 2
**Presentation:** 1
**Contribution:** 2
**Rating:** 3
**Confidence:** 4

**Summary:**

In this paper, the authors introduces the Weight Augmentation Strategy (WAS), it aims to enhance the robustness and adaptability of deep neural networks by modifying weights rather than focusing on traditional network structures or data augmentation. WAS involves using plain weights (PW) and shadow weights (SW), where PW represents the original weights, SW are randomly transformed versions of PW that collectively from a high-dimensional weight space during training. Besides, the authors introduced two operational modes, the Accuracy-Oriented Mode (AOM) and Desire-Oriented Mode (DOM), where AOM utilises PW to maintain robustness and accuracy, DOM utilises SW to optimise for specific tasks, e.g., reducing computation.

**Strengths:**

1. The concept of extending augmentation from data to weights by leveraging the WAS is somewhat novel, it brings a fresh perspective by exploring the distribution of weights spaces and using transformations to enhance generalisation, which hasn’t been widely explored in the previous work.

2. The dual-mode operation provides flexibility for both robust performance and reduced computation.

3. DOM showed notable reductions in FLOPs which benefiting models under resource constraints. DOM is able to effectively handle extreme data manipulations, even better than AOM in some cases.

**Weaknesses:**

1. The writing is a bit poor, contains many typos, e.g., in line 20, “the Accuracy-Priented Mode (AOM)”. As well as some over-claims, e.g.,  in line 28, the authors claim that WAS can “significantly enhance almost all networks”, but only convolutional neural networks been explored in this paper. How about ViT? The authors can consider either rephrasing the description or adding more evidence to support this claim.

2. Although the experiments show some improvements, they are limited to CIFAR-10 and CIFAR-100. It would be more convincing if the authors can include experiments on more diverse and challenging datasets like ImageNet, as well as other types of neural networks such as Transformers.

3. While the empirical results look strong, the theoretical discussion of why shadow weights work effectively is inadequate. A more thoroughly investigation and analysis of how WAS contributes to better generalisation from a theoretical perspective would ass more depth to the understanding of this method.

**Questions:**

1. How to decide when to switch between AOM and DOM during inference? Depends on what criteria or heuristics? This is crucial as the feasibility of deploying a model with two operational modes largely depends on how effectively it can switch modes autonomously.

2. While DOM reduces FLOPs during inference, can authors provide more detailed insights regarding how WAS affects overall training time and computational resources? Are there any overheads due to the generation of the shadow weights?

3. The authors currently use random transformations like rotation, scaling and cropping for the weight augmentation, have you explored other types of transformations for generating the shadow weights? If so, how do they behave in terms of model performance?

---

### Official Review · Reviewer_ELTM · 2024-11-02

**Soundness:** 2
**Presentation:** 1
**Contribution:** 2
**Rating:** 3
**Confidence:** 3

**Summary:**

This work introduces a weight augmentation mechanism for training CNNs. The proposed method can operate in two different modes, that can either boost performance, or focus on more specific objectives like reducing the computational complexity. WAS can be employed in a variety of CNN architectures. The experiments show that, depending on the chosen mode, the proposed method can significantly boost performance, or reduce the number of FLOPs.

**Strengths:**

The idea of weight augmentation is quite interesting and novel in the context of training CNNs.

The experiments demonstrate that WAS can be used to increase model performance on a multitude of CNNs, or to reduce computational complexity, depending on the needs of the tasks at hand.

**Weaknesses:**

The writing of the manuscript should be heavily improved, it is often unclear and not very coherent.
More specifically, the introduction does not flow nicely; new terms (including PW and SW) are not properly explained, the connections with other works (e.g. Dropout) are not so clear and are only mentioned briefly.
The related work section is extremely short and does not seem to place this work well within the literature.
Furthermore, the method section is lacking in thoroughness and clarity. See questions 1-4 for more details.
Finally, the experiment section focuses heavily on the relative performance gap between the method and baselines, and less on the intuitive explanations as to why that is the case.

The authors only run experiments on two datasets, CIFAR10 and CIFAR100, which are also very small scale datasets for modern deep learning. Do the conclusions drawn from this work hold on ImageNet-level datasets? Do they diminish with more training data?

As a minor note on the writing, named citations are often mentioned twice, while parentheses are lacking in the remaining citations.

**Questions:**

[1] In equations 3-4, the components are not explained thoroughly. What do $h$, $k$, $T$, $x$, $b$, $W$ denote? What are their dimensions?

[2] What does the term "each weight component" refer to in line 192? Is it each row of the weight matrix? Is $W$ a convolutional kernel or a weight matrix?

[3] It is unclear from the manuscript if the geometric transformations from equation 3 are part of the related work, or if they are proposed by this work. Is this the first work to apply geometric transformations in the weight space?

[4] In equation 3, is T a 2D geometric transformation, or is it a transformation on the hidden dimension of the network? If it is a 2D transformation, shouldn't the equation be $W T x$ instead? Is the transformation applied to the weights of the first layer only?

[5] The paper claims that WAS can enhance almost all methods, however, the experiments only focus on CNNs. Is the method applicable to other architectures, e.g. Vision Transformers?

[6] In line 423, the authors mention that they "conducted random data augmentation on the test set". What is the purpose of this augmentation?

---

### Official Review · Reviewer_YMQo · 2024-11-02

**Soundness:** 2
**Presentation:** 1
**Contribution:** 2
**Rating:** 3
**Confidence:** 5

**Summary:**

The work in this paper introduces the Weight Augmentation Strategy (WAS), a novel training approach that shifts focus from optimizing network structures to exploring weight distributions.

**Strengths:**

1. WAS provides significant accuracy improvements across multiple architectures on CIFAR-10 and CIFAR-100 datasets.
2. Table 2 shows the computational cost is reduced significantly by the proposed method.

**Weaknesses:**

1. Writing needs to be improved. There are many typos, for example:
(1) At the beginning of the introduction, "learning" should be "Learning".
(2) Some "et al."s are in italics while some are not.
(3) Some equations have commas at the end while some do not.
(4) In Eq. (5), should $J(\theta)$ be $J(\theta_j)$? is there a typo?

2. This work is hard to follow:
(1) It is confusing why Eq. (1) is mapping from weights to the input tensor.
(2) In Eq. (4), j is not defined.

3. The networks in this work are very old. Only convolutional neural networks up to the year 2019 are discussed. The import transformers or ConvNext are missing.

4. Experiments are limited. The authors only made evaluations on the CIFAR-10/100 classification tasks. Other important benchmarks such as ImageNet-1K classification and COCO-2017 object detection are missing.

4. It is not clear why the authors discuss different data augmentation methods in Table 1. In Figure 1, the authors mentioned that their novelty is training the weight distribution instead of weights themself. However, these data augmentations (rotation, cropping, translation) are very common in the classification tasks.

**Questions:**

1. In Eq. (5), should $J(\theta)$ be $J(\theta_j)$? is there a typo?
2. It is not clear why the authors discuss different data augmentation methods in Table 1. In Figure 1, the authors mentioned that their novelty is training the weight distribution instead of weights themself. However, these data augmentations (rotation, cropping, translation) are very common in the classification tasks.

---

### Official Review · Reviewer_pfZh · 2024-11-03

**Soundness:** 2
**Presentation:** 1
**Contribution:** 3
**Rating:** 5
**Confidence:** 3

**Summary:**

This paper introduces a weight augmentation strategy (WAS) designed to improve model robustness by transforming model parameters (weight augmentation) rather than applying data augmentation during training. The authors propose two operational modes: the Accuracy-Oriented Mode (AOM) and the Desire-Oriented Mode (DOM). AOM aims to maintain high robustness and accuracy, while DOM focuses on specific objectives, such as reducing computational complexity or minimizing sensitivity to particular types of data. Experiments were conducted using CIFAR-10 and CIFAR-100 datasets with models including VGG-16, ResNet-18, ResNet-34, GoogleNet, MobileNetV2, and EfficientNet-Lite.

**Strengths:**

1， This paper proposes a novel perspective point to enhance the robustness of the model, by augmenting the weight instead of data.

2,    Conduct experiments on several models (VGG-16, ResNet-18, ResNet-34, GoogleNet, MobileNetV2, and EfficientNet-Lite).

**Weaknesses:**

1, the paper is a very first draft, containing lots of typos and unclear claims,  such as: the symbol of partial derivation in line 227; in line 340-350, maybe the description of Table 1 is not right; what does the drop rate mean in the table? ; how do you calculate FLOPs?


2, the paper lacks key details about their proposed method, such as:  how you select the hyperparameter for the degree of randomness in the transformation; and how exactly do the transformation for the weight (this is your key operation for your paper).


3, the structure of the paper is not very good, making it kind of hard to follow.

**Questions:**

1 please explain how the method facilitates learning data distribution in line 238-239.

2 please explain: "The WAS incentive model integrates hundreds of weight training iterations, leveraging the advantages of SW while mitigating potential drawbacks."

3 the paper states multiple times that DOM  can reduce the computation. could you explain how it can be reduced?  As I understand, when you crop the weight, maybe you will keep some weight to zero? how every is will attend computation, right?

---

### Meta-Review · Area_Chair_73Na · 2024-12-21

**Metareview:**

The paper proposes an interesting concept of weight augmentation by random transformations, reminiscing of a more Bayesian treatment of neural networks. While interesting, all reviewers agree that the paper is lacking in presentation and clarity, specifically with regards to key concepts of the paper. The authors did not submit any rebuttal, so the paper should anyway go for another round of revision.

**Additional Comments On Reviewer Discussion:**

There was no discussion since there was also no rebuttal.

---

### Decision · Program_Chairs · 2025-01-22

Reject